# Isolation and Selection of Protein-Rich Mutants of *Chlorella vulgaris* by Fluorescence-Activated Cell Sorting with Enhanced Biostimulant Activity to Germinate Garden Cress Seeds

**DOI:** 10.3390/plants13172441

**Published:** 2024-09-01

**Authors:** Mafalda Trovão, Lisa Schüler, Humberto Pedroso, Ana Reis, Gonçalo Espírito Santo, Ana Barros, Nádia Correia, Joana Ribeiro, Gabriel Bombo, Florinda Gama, Catarina Viana, Monya M. Costa, Sara Ferreira, Helena Cardoso, João Varela, Joana Silva, Filomena Freitas, Hugo Pereira

**Affiliations:** 1Allmicroalgae Natural Products S.A., R&D Department, 2445-413 Pataias, Portugal; mafalda.trovao@allmicroalgae.com (M.T.); humberto.pedroso@allmicroalgae.com (H.P.); anaisabel_1995@hotmail.com (A.R.); moisesgoncalo.16@gmail.com (G.E.S.); ana.barros@allmicroalgae.com (A.B.); nadia.correia@allmicroalgae.com (N.C.); joanaribeiro.18@gmail.com (J.R.); helena.cardoso@allmicroalgae.com (H.C.); 2GreenCoLab, Associação Oceano Verde, University of Algarve, 8005-139 Faro, Portugal; lisa.schueler@quazyfoods.com (L.S.); gabrielbombo@greencolab.com (G.B.); florindagama@greencolab.com (F.G.); catarinaviana@greencolab.com (C.V.); monyacosta@greencolab.com (M.M.C.); saraferreira@greencolab.com (S.F.); jvarela@ualg.pt (J.V.); jglaranjeira2@gmail.com (J.S.); 3Associate Laboratory i4HB, Institute for Health and Bioeconomy, School of Science and Technology, NOVA University Lisbon, Campus de Caparica, 2829-516 Caparica, Portugal; a4406@fct.unl.pt; 4UCIBIO-Applied Molecular Biosciences Unit, Department of Chemistry, School of Science and Technology, NOVA University Lisbon, Campus de Caparica, 2829-516 Caparica, Portugal; 5CCMAR, Centre of Marine Sciences, University of Algarve, 8005-139 Faro, Portugal

**Keywords:** biostimulants, fluorescence-activated cell sorting, microalgae, protein, random mutagenesis, selection method

## Abstract

Microalgae are a promising feedstock with proven biostimulant activity that is enhanced by their biochemical components (e.g., amino acids and phytohormones), which turns them into an appealing feedstock to reduce the use of fertilisers in agriculture and improve crop productivity and resilience. Thus, this work aimed to isolate protein-rich microalgal mutants with increased biostimulant activity. Random mutagenesis was performed with *Chlorella vulgaris,* and a selection of protein-rich mutants were sorted through fluorescence-activated cell sorting (FACS), resulting in the isolation of 17 protein-rich mutant strains with protein contents 19–34% higher than that of the wildtype (WT). Furthermore, mutant F4 displayed a 38%, 22% and 62% higher biomass productivity, growth rate and chlorophyll content, respectively. This mutant was then scaled up to a 7 L benchtop reactor to produce biomass and evaluate the biostimulant potential of this novel strain towards garden cress seeds. Compared to water (control), the germination index and the relative total growth increased by 7% and 19%, respectively, after the application of 0.1 g L^−1^ of this bioproduct, which highlights its biostimulant potential.

## 1. Introduction

One in three people struggle with moderate to severe food insecurity worldwide [1]. Nowadays, food and feed production do not meet global demands for these commodities, so more productive and resistant crops are required [2]. The abusive usage of chemical fertilisers, pharmaceuticals, antibiotics and pesticides, either in agriculture or in meat/fish production, contributes to a build-up of water and land contamination as well as eutrophication, pest resistance and increased incidence of human illnesses [3,4,5,6].

Plant biostimulants are an important resource for shifting traditional agricultural practices into more sustainable, safe, and productive processes. These compounds and/or extracts enhance plant growth by improving nutrient uptake, root development and crop resilience while reducing the reliance on chemical fertilisers and synthetic pesticides [2]. Microalgae are an under-exploited resource that are of interest either as a feedstock to develop food products and feed formulations or as a product to address the sustainability of existing processes, namely by bioremediation and by their potential as biopesticides, biostimulants and immunity-boosters, both in agriculture and aquaculture [7,8,9,10].

Microalgal biocompounds, such as amino acids, polysaccharides, phenolic compounds and minerals, and phytohormones, namely auxins, cytokinins, ethylene and gibberellins, have been reported as potent biostimulants and/or biopesticides to improve crop performance, quality and stress tolerance [11,12,13,14,15]. In addition, *C. vulgaris* has been reported as one of the dominating species with biostimulatory activity [12,13,16]. Particularly, protein- and amino acid-based biostimulant application has been reported to have many positive effects on several plant species, as reviewed by Sun et al. [17]. In addition, molecules with high antioxidant potential, such as chlorophyll and carotenoids, have protective effects that might contribute to plants’ enhanced growth [18]. The anti-oxidative, anti-microbial and immunomodulatory activities of these compounds promote seed germination, alleviate the impact of environmental stress factors, such as high salinity, drought and contaminants, boost crops’ productivity and might decrease the need to apply chemical fertilisers, namely by playing a role as osmolytes, aid in heavy metal detoxification, increase plants’ absorption of soil micronutrients and improve the enzymatic antioxidant defence machinery of plant cells [17].

Notwithstanding, many microalgal wildtype (WT) strains often do not present the desired traits for industrial production, either in terms of growth performance, lack of robustness, colour or low target compound content [19]. For that reason, several approaches have been used lately to improve wildtype strains, namely adaptive laboratory evolution, random mutagenesis and genetic engineering. Random mutagenesis is a well-established technology with fast-result delivery, in which cells are exposed to a physical or chemical agent that generates random mutations in the genome [19,20]. In addition, the strains generated are not subjected to the restrictions imposed on genetically modified organisms since no foreign genetic material is introduced into the cells [19,21]. Although a vast library of mutants might be obtained, it is very laborious and time-consuming to identify and select the phenotypes of interest. Up until now, few methods have been developed in this regard, namely the use of metabolic pathway inhibitors, visual appearance or autofluorescence of colonies and fluorescence-activated cell sorting (FACS) [19]. Moreover, the few high-throughput strategies available have been developed mostly for the selection of mutants with faster growth, higher lipid content and improved pigment content [22,23,24,25,26]. Thus, there is a scarcity of selection strategies for other target compounds of interest, such as proteins and amino acids.

In this context, a novel approach was attempted in this study. According to Malerba et al. [27], it is possible to establish a correlation between standard flow cytometric properties, such as side scatter (SSC), forward scatter (FSC) and red fluorescence, derived from the pigments (chlorophyll) of microalgal cells and cell nitrogen quota. Based on the model by these authors, it was hypothesised that cell nitrogen quota would also have a correlation with cell protein content. Therefore, cells of a larger size (as estimated by FSC), higher complexity (as estimated by SSC) and higher chlorophyll content (≈higher red autofluorescence) were selected in this work using FACS [28].

*Chlorella vulgaris* is one of the few industrial species of microalgae with the ability to grow heterotrophically, which contributes to the fact that some of the highest biomass productivities reported for microalgae cultivation were achieved with this species [29,30]. Heterotrophic cultivation has a smaller areal footprint than photoautotrophic cultivation, requires less land and water usage and is independent of climatic conditions, allowing for the consistent achievement of significantly higher cell concentrations [31,32,33,34]. Moreover, this species accumulates high protein contents that can go from 20 to 64% of dry weight (DW) [35,36].

In this work, a novel selection strategy that resorts to FACS was established to isolate protein-rich mutants of *C. vulgaris*. The mutants generated were characterised and compared between them and with the wildtype. Based on the growth performance and protein and pigment content, the most promising mutant was selected to be scaled up in a 7-L benchtop fermenter for biomass production. Finally, the biostimulant activity of the produced biomass was compared to that of a commercial algae-based biostimulant and the phytohormone gibberellic acid (GA) in germination trials of garden cress seeds.

## 2. Results

### 2.1. Mutagenesis and Isolation of Protein-Rich Mutants by FACS

The mutagenised cells of *C. vulgaris* were acquired in the cytometer after the recovery period. To select mutants with higher protein and chlorophyll contents, two gates were set, P1 and P2 (Figure 1).

The first gate, P1, was set to a higher side scatter (SSC) signal, while the second gate, P2, was set to a higher forward scatter (FSC) and a higher chlorophyll autofluorescence (PerCP-Cy5-5-A) signal. This fluorescence-activated cell sorting (FACS) procedure allowed the isolation of 17 mutants of *C. vulgaris* with increased cell complexity, larger cell size/volume and higher chlorophyll autofluorescence.

### 2.2. Screening of Mutants

#### 2.2.1. Growth Performance and Protein Content

The 17 mutants isolated by FACS were named following the respective well position (in the 96-well plate) to which they were sorted. The growth performance of these mutants was compared with that of the wildtype (WT). Their respective growth parameters, biomass productivity and growth rate, as well as their protein content, are shown in Table 1.

From the 17 mutants, only mutant E2 displayed a significantly lower biomass productivity and growth rate (0.84 ± 0.04 g L^−1^ d^−1^ and 0.95 ± 0.01 d^−1^) when compared to the WT (1.51 ± 0.05 g L^−1^ d^−1^ and 1.05 ± 0.02 d^−1^). On the other hand, three mutants, F4, F5 and G2, exhibited higher growth rates (1.23–1.28 d^−1^) than the WT, but only mutant F4 presented significantly higher biomass productivity (2.08 ± 0.17 g L^−1^ d^−1^), which corresponded to a 38% increase. All mutants displayed protein contents that averaged 31% of their dry weight (DW), which was similar to that of the WT (32.14 ± 1.07% DW). However, the mutants C5, E2, F4, F5 and G2 displayed significantly higher protein contents (38.05–42.98% DW), which represented an improvement of 19–34% as compared to the WT. Regarding protein productivity, only mutant E4 (0.39 ± 0.01 g L^−1^ d^−1^) exhibited a significantly lower value compared to that of the WT (0.56 ± 0.05 g L^−1^ d^−1^). The highest values were achieved by mutants C4, C5, F4, F5, G2 and G3, between 0.60 and 0.68 g L^−1^ d^−1^. The growth curves of the mutants mentioned above are shown in Figure A1 (Appendix A).

#### 2.2.2. Chlorophyll and Carotenoid Profiles

The pigment profiles of the WT and the mutants that displayed improved growth performance and protein content, F4 and G2, were analysed (Table 2).

Regarding the pigments’ profile, the WT strain presented lower total chlorophyll (0.48 ± 0.05 mg g^−1^ of DW) and lutein (0.53 ± 0.01 mg g^−1^ of DW) contents when compared to the mutant strains F4 and G2. Mutant strain F4 displayed the highest chlorophyll content (0.78 ± 0.01 mg g^-1^ of DW), a 62% increase compared to the WT. Additionally, mutant G2 exhibited a 55% increase in lutein content (0.82 ± 0.08 mg g^−1^ of DW), as compared to the WT.

### 2.3. Biomass Growth and Protein Production

Mutant strain F4 exhibited the best growing performance compared to the other mutants, as well as the most interesting pigment profile, along with a 19% improvement in protein content. Thus, F4 was selected for scale-up in a 7-L benchtop reactor to generate enough biomass to assay its biostimulant potential (Figure 2).

Mutant F4 reached 81.9 ± 18.2 g L^−1^ of DW in 7 days in the 7-L reactor, with a biomass productivity of 13.9 ± 3.6 g L^−1^ d^−1^, and a specific growth rate of 0.66 ± 0.13 d^−1^ (Figure 2). The final protein content attained was 33.1 ± 1.2% of DW, which comes with a protein productivity of 3.9 ± 0.7 g L^−1^ d^−1^ (Figure 2). In the first 2 days of cultivation, a lag phase was observed, a period that could potentially be shortened through process optimisation, along with an increase in growth rate and protein content, similar to what was achieved in the laboratory screening trials.

### 2.4. Biostimulant Activity (In Vitro Assays)

The whole biomass of mutant F4 obtained in the 7-L fermenter was applied to garden cress seeds, whose germination index (GI), relative radicle growth (RRG) and relative total growth (RTG) were compared to those of water, a commercial algae-based biostimulant (Algaman) and gibberellic acid (GA), as shown in Figure 3.

Regarding the positive controls, GA increased the germination by 20%. However, Algaman led to a 30% decrease in the GI. A similar behaviour was obtained for RRG and RTG. Treatments with 0.01, 0.1 and 0.25 g L^−1^ of the F4 strain biomass enabled the same response as the GA for the three parameters measured, with an improvement in RRG between 3 and 8%, RTG between 13 and 19% and GI between 3 and 8%, as compared to those of water. Concentrations equal to or higher than 0.5 g L^−1^ of microalgal biomass significantly impaired the three parameters under study concerning the biostimulant activity on garden cress seeds.

## 3. Discussion

The optical properties detected by flow cytometers have been extensively correlated with cell features across different microalgae and cyanobacteria species [24]. Although it is an oversimplification, the forward scatter (FSC) detector is often considered proportional to cell size/volume. In contrast, side scatter (SSC) is considered a good proxy for cell internal complexity/granularity. The red (auto)fluorescence signal is proportionally correlated to the total pigment concentration of the cells, particularly chlorophyll [24,27,37]. In addition, several strategies that resort to fluorescent dyes have been used to identify and isolate species with higher contents of high-value compounds, such as lipids and carotenoids [19,24]. However, no strategy has been developed to distinguish and measure the protein content of living cells. Usually, the protein content of microalgae is quantified by indirect or direct methods, either by quantifying the nitrogen content, for example by elemental analysis [38], or by digestion protocols and colorimetric reactions such as Kjeldahl’s [39], Lowry’s [40] and Bradford’s [41] methods, which are often inaccurate and time-consuming [27]. Unlike lipids, which can be stained, for example, with the solvatochromic dye BODIPY [42], there is no standard procedure to stain proteins without compromising the cell viability of the microalgae and disrupting cell membranes.

Nonetheless, Malerba et al. [27] reported a method that correlates the optical properties of flow cytometry with cell nitrogen quota to monitor phytoplankton populations, since nitrogen limitation is known to affect the physiological and morphological aspects of cells. These authors established a model with high accuracy (*R^2^* = 0.9; Prob (F) < 0.0001) whose most important variable was red fluorescence, which explained 77% of the variability of the total cell nitrogen, which increased to 87% when combined with SSC and went up to 90% when also combined with FSC. This method allowed them to establish a quantifiable proxy for cell nitrogen quota in a reliable and non-destructive manner across four species (*Desmodesmus armatus*, *Mesotaenium* sp., *Scenedesmus obliquus* and *Tetraëdron* sp.). Based on this model and this correlation with optical properties, it was hypothesised that the mutants isolated with a higher cell nitrogen quota (Figure 1) would have a higher protein content.

The protein-rich mutants isolated by FACS were compared at laboratory scale (Table 1). At this scale, most reports found in the literature for *C. vulgaris* cultivated in heterotrophic conditions presented lower growth rates, between 0.55 and 0.79 day^−1^, compared to all the strains screened here, while biomass productivities fell within a similar range, 1.65–1.99 g L^−1^ d^−1^ [35,36,43,44]. However, upon random mutagenesis, both impaired and improved growth performances of the generated mutants have been reported, depending on the improvement target, mutagen and selection method used. For example, Schüler et al. [20] developed a yellow mutant of *C. vulgaris* that displayed a growth performance equivalent to the WT and a white mutant that grew slower than the WT. The yellow and white strains showed a 30 and 60% increase in protein content, respectively (39.5 and 48.8% of dry weight (DW), respectively), as compared to the WT, even though that was not the primary target of the mutagenesis. Conversely, aiming for different improvement targets, several reports of improvements in the growth performance of several microalgal species after mutagenesis have been published [45]. Kim et al. [46] reported a 1.3-fold improvement in the growth rate of *C. vulgaris* after combining mutagenesis with ethyl methanesulfonate (EMS) with FACS-based selection in order to improve carotenoid content. Improved growth performance and the biodegradative potential of petroleum was also reported by Eregie et al. [47] by applying UV-radiation to mutagenise *Scenedesmus vacuolatus*, which also led to a 2-fold increase in chlorophyll content, a 1.2-fold increase in carotenoids and 1.4-fold increase in the protein content. In addition, Liu et al. [48] generated an *Auxenochlorella pyrenoidosa* mutant by atmospheric room temperature plasma mutagenesis with a 31% increase in protein content (44.22% DW), with no detectable chlorophyll *b* and a 118-fold decrease in chlorophyll *a* content, comparing to the WT, without significant differences regarding growth performance.

The chlorophyll contents obtained in this study (Table 2) were significantly lower than other values reported for this species under heterotrophic conditions [20,30,45], which might be related to the strain used and/or the efficiency of the extraction. In resemblance to growth performance, chlorophyll and carotenoid enhancements and decays have both been reported upon mutagenesis, also depending on the objective [49,50,51,52,53,54]. Chlorophyll-deficient mutants in some reports displayed improved growth performance, and were impaired in others [45,49,55]. Regarding chlorophyll increments, Nakanishi and Deuchi [56] presented a three-fold increase in chlorophyll content, along with increased halotolerance, upon the UV-mutagenesis of *C. vulgaris* and selection by colour of the colonies generated. By resorting to UV-mutagenesis, Vigeolas et al. [57] also isolated a mutant of *Tetradesmus obliquus* with a 2.2-fold higher chlorophyll and protein content, but with an impaired doubling rate, upon Nile red fluorescence-based screening for cells with higher lipidic contents. In addition, Xi et al. [58] also generated mutants of *T. obliquus* by ^12^C^6+^ Ion Beam mutagenesis and selected a strain with 33% and 48% higher chlorophyll *a* and carotenoids contents, respectively, through chlorophyll fluorescence, even though improving photosynthetic efficiency and lipid content were the original goals. Furthermore, random mutagenesis can alter the carotenoids profile, as Kim et al. [46] demonstrated by generating a mutant of *C. vulgaris* to accumulate violaxanthin.

The growth performance of the F4 strain in the 7-L reactor (Figure 2) achieved a biomass productivity of 13.9 g L^−1^ d^−1^, which is higher than most values that have been reported for this species, between 1.7 and 3.2 g L^−1^ d^−1^, [20,35,36,43,44]. Still, it falls short of the value reported by Barros et al. [30], namely 27.3 g L^−1^ d^−1^. In addition, the protein content achieved, 33.1% DW, was lower than the value obtained in the screening assay, 38.1% DW (Table 1). In the literature, lower and higher protein content values are found, ranging between 20 and 64% of DW [35,36], as reviewed recently by Trovão et al. [45]. However, the F4 strain’s protein productivity is higher than most values reported due to the high biomass productivity obtained. It is also noteworthy that both growth performance and protein productivity still have a great margin of improvement, which would require the optimisation of the culture medium, abiotic factors, cultivation and feeding mode, as well as eventually performing a two-stage process to enhance protein production [59,60]. Furthermore, higher productivities, both of biomass and target compounds, as well as growth rates, might be achieved upon the scaling-up of the process, as it has been pointed out for *C. vulgaris* and other species, such as *S. rubescens* [30,61].

Regarding the biostimulant activity assays (Figure 3), similar results were reported by Morillas-España et al. [62], which attained an increment of 3.5% of the GI of watercress seeds also when using 0.1 g L^−1^ of *C. vulgaris*, highlighting the biostimulant capacity of this species. However, these authors applied this microalgal extract after cell wall disruption by sonication, while in the present study, non-disrupted biomass was applied directly instead. In addition, these authors also reported the promotion of root formation in soybean seeds, a cytokinin-like effect in a cucumber expansion test and the formation of chlorophyll in wheat leaves after treatments with *C. vulgaris* extract. On the other hand, Gitau et al. [63] treated a *Medicago truncatula* model plant with live algae cells of *Chlorella*, which led to larger leaves, more flowers/pods, increased fresh biomass and more robust plants compared to the control. Alling et al. [64] tested the biostimulating effects of both the algal biomass (intact vs. disrupted cells) and supernatant (after cultivation) of *C. vulgaris* on tomato and barley seeds. Intact cells and their supernatant enabled up to a 25% higher germination percentage, higher GI and earlier germination by 0.5–1 day when evaluated against seeds treated with *S. obliquus* and the negative control (water). Martini et al. [65] also reported improved development of maize roots when plants were treated with *C. sorokiniana*, compared to the untreated negative control, under stress conditions, such as nitrogen depletion. In addition, these authors suggested that the absence of pretreatment of the biomass enables the establishment of a more sustainable process, since the physical treatment of cells that they performed (partial disruption with glass beads before freeze-drying) had a limited effect on their biostimulant properties compared to the untreated freeze-dried biomass. Finally, Gharib et al. [66] recently reported the impact of microalgal extracts (obtained through methanol extraction, solvent evaporation and resuspension in water) of several species, including *C. vulgaris*, as biostimulants on common bean plant growth, yield and antioxidant capacity. The most promising results were obtained with extract concentrations between 0.5 and 1.0%, which improved root and shoot length, number and area of leaves, weight per plant, seed index and yield per plant, as well as reduced content of oxidative stress markers, among other positive effects. Besides the biostimulant potential of aqueous suspensions of *C. vulgaris*, other effects have been reported recently, namely as a biocontrol agent/biopesticide against *Fusarium oxysporum* to protect spinach [67].

Although the beneficial effect of microalgal biomass as a biostimulant and bioprotective agent has been reported by several authors, as reviewed by Mrid et al. [68], it would be interesting to study further which compounds provide these effects and the underlying mechanisms, namely by identifying and quantifying phenolic compounds, phytohormones, amino acids and polysaccharides, for example. While some of the mechanisms of these compounds’ biostimulant activity has been reported, others have not been investigated comprehensively. For example, pigments, such as carotenoids, are precursors of known phytohormones, such as strigolactones and abscisic acid [69,70]. It would be worth studying the effect of these molecules on plants’ growth and resistance to stress.

Finally, most of these bioactive microalgal compounds are intracellular, and *Chlorella* spp. and other species are known for having a recalcitrant cell wall. Although there are already a few studies comparing the usage of intact vs. disrupted cells or even supernatant, as discussed above, it would be important to further study the effect of different biomass treatments on the composition of microalgal extracts for this application. Such treatments could include different disruption methodologies, namely high-pressure homogenisation, pulsed electric fields and enzymatic and acid or alkali hydrolysis [71,72]. In addition, the conditions applied in these processes must be optimised to avoid compromising the bioactivity of the target compounds.

## 4. Materials and Methods

### 4.1. Microalgae Strain and Mutagenesis

*Chlorella vulgaris* wildtype strain 8 (8WT) was obtained from the Allmicroalgae Natural Products S.A. culture collection from cryopreserved aliquots stored in liquid nitrogen (−196 °C). The heterotrophic medium (HM) described by Barros et al. [30] was used to cultivate this strain, with 30 mM ammonium sulphate and 20 g L^−1^ glucose.

The dose-response curve of ethyl methanesulfonate (EMS) was established by exposing the 8WT strain to concentrations from 0 to 300 mM of EMS (1.90 × 10^8^ cells mL^−1^) according to the protocol described by Trovão et al. [45]. In order to maximise the generation of strains with single mutations in their genomes, a concentration of 200 mM of EMS was selected for further rounds of mutagenesis, allowing close to a 10% survival rate obtained when different concentrations of EMS were assayed (Figure 4). Upon mutagenesis, cells were recovered by resuspending in the respective diluted HM medium (1:2) and incubated overnight.

### 4.2. Isolation of Protein-Rich Strains by FACS

The fluorescence-activated cell sorting (FACS)-based screening procedure performed was based on the model reported by Malerba et al. [27] that correlates flow cytometric properties (red fluorescence, side scatter (SSC) and forward scatter (FSC)) with intracellular nitrogen quota. The mutagenised *C. vulgaris* cells were acquired in a Becton Dickinson FACS Aria II (BD Biosciences, Erembodegem, Belgium) equipped with a blue, violet and red laser and FACSDiva (version 6.1.3) software. The fluorescence signal of chlorophyll was obtained by applying a filter at 695/40 nm after excitation with the blue laser (488 nm). Cells were gated first for higher complexity (SSC) and then for those emitting higher levels of fluorescence and higher size (FSC), which were sorted onto 96-well microplates containing 250 µL of HM medium and incubated at 30 °C. From the wells that presented cell growth after 15 days, cultures were transferred to Petri dishes containing plate-count agar (PCA). Both the microplates and PCA plates were incubated in the dark at 30 °C.

### 4.3. Screening of FACS Mutants

#### 4.3.1. Growth Performance

The mutants isolated on PCA plates were then transferred to 250-mL Erlenmeyer flasks with 50 mL of HM medium to compare their growth performance with that of the wildtype strain. *C. vulgaris* 8WT and FACS-selected mutants were cultivated at 30 °C in an orbital incubator (ArgoLab^®^ shaker SKI 4, Capri, Italy) at 200 rpm. PIPES buffer at 50 mM was added to the medium to keep the pH at 6.5.

Growth was followed by measuring the optical density at 600 nm (OD600) in a spectrophotometer (Genesys 10S UV-Vis^®^; Thermo Fisher Scientific, Massachusetts, USA), optical microscopy observation (Axio Scope A1^®^, Carl Zeiss Microscopy GmbH, Oberkochen, Germany) and pH measurements (Metria universal pH test paper strips; Labbox Labware, SL, Barcelona, Spain). Samples were dried and weighed in a moisture analyser (MA 50.R Moisture Analyser, Radwag^®^, Radom, Poland) at 120 °C to determine their dry weight (DW) after pre-weighing the filters (0.7 μm glass microfibre; VWR International, Philadelphia, USA) and washing the microalgal suspensions with demineralised water. DW was calculated by the following equation:(1)$$DW\left(gL^{-1}\right)=\frac{\left(m_{f}-m_{i}\right)}{V}$$
where *m_i_* corresponds to the mass of the filter, *m_f_* to the mass of the filter plus the algal biomass collected in it and *V* to the volume of culture filtrated.

The correlations established between OD600 and the dry weight (DW) of *C. vulgaris* 8WT (Equation (2); *R*^2^ = 0.992) and the established mutant 8F4 (Equation (3); *R*^2^ = 0.976) were the following:(2)$$OD\left(600\mathrm{nm}\right)=\left(DW/0.4255\right),$$
(3)$$OD\left(600\mathrm{nm}\right)=\left(2.6258\times DW\right),$$

The biomass productivity (*P*) and growth rate (*µ*) were calculated by Equations (4) and (5), where *DW_i_* and *DW_f_* correspond to the final and initial dry weights measured at the beginning of the assays (*t_i_*) and at the end of the exponential phase (*t_f_*), respectively:(4)$$\begin{array}{c} P(gL^{-1}{day}^{-1})=\frac{\left({DW}_{f}-{DW}_{i}\right)}{\left(t_{f}-t_{i}\right)} \end{array},$$
(5)$$\begin{array}{c} \mu ({day}^{-1})=\frac{\ln({DW}_{f}/{DW}_{i})}{\left(t_{f}-t_{i}\right)} \end{array},$$

Finally, samples were collected by centrifugation at 4500× *g* for 15 min (Hermle^®^ Z300 centrifuge, Gosheim, Germany), and the biomass was frozen at −20 °C.

#### 4.3.2. Biochemical Analysis of the Biomass

The frozen samples stored previously were lyophilised in a Coolvacuum, Lyomicron (Barcelona, Spain), and stored in a desiccator for the quantification of protein and pigments’ contents at a later step.

##### Protein Content

Protein content was calculated by multiplying the nitrogen content by a 6.25 factor [38]. The nitrogen content was determined by elemental analysis (Vario EL III^®^, Elemental Analyser System; GmbH, Hanau, Germany) according to the manufacturers’ instructions.

##### Extraction and Quantification of Chlorophyll and Carotenoids

Chlorophyll was extracted and quantified based on Ritchie’s method [73]. For that purpose, 10 mg of biomass plus 2 g of glass beads (dp = 1 mm) and 6 mL of acetone (99%) were added to a glass tube and vortexed for 10 min. To separate the extract from the cells, the samples were centrifuged for 10 min at 2547× *g* (Hermle^®^ Z 300 centrifuge, Wehingen, Germany). After centrifugation, the supernatant was collected and kept in the dark at −20 °C. This extraction step was repeated until the biomass became colourless.

Finally, chlorophyll *a* and *b* were quantified by the absorbance of the supernatant of the samples at 630, 647, 664 and 691 nm, followed by calculating the concentrations as indicated in Equations (6) and (7):(6)$$\begin{array}{c} {Chl}_{a}=-0.3319{Abs}_{630}-1.7485{Abs}_{647}+11.9442{Abs}_{664}-1.4306{Abs}_{691} \end{array}$$(7)$$\begin{array}{c} {Chl}_{b}=-1.2825{Abs}_{630}+19.8839{Abs}_{647}-4.8860{Abs}_{664}-2.3416{Abs}_{691} \end{array}$$

Carotenoids were extracted and quantified according to the protocol reported by Trovão et al. [45]. Briefly, 5–10 mg of biomass was weighed, and 1 mL of methanol containing 0.03% butylhydroxytoluene was added to each sample, as well as 0.6 g of glass beads (dp~425–600 μm). After bead milling for 3 min at 30 Hz with a mixer mill (Retsch MM 400, Vila Nova de Gaia, Portugal), samples were centrifuged for 3 min at 24,000× *g,* and the supernatant was collected after that. As for chlorophyll extraction, this step was repeated until the biomass and the supernatant became colourless. After evaporating the extracts under continuous nitrogen flow, the dried samples were resuspended in 1 mL HPLC grade methanol and filtered through a 0.22 μm PTFE filter.

The quantification of the carotenoids of each sample was carried out with a Chromaster HPLC System (Hitachi, VWR, Carnaxide, Portugal), equipped with a diode array detector (5430 DAD, Hitachi, VWR, Carnaxide, Portugal) and a Purospher^®^ STAR RP-18 endcapped (Merck, Portugal) (250 × 2.1 mm, 5 μm) chromatographic column. The following gradient of solvent A (acetonitrile:water 9:1, *v*/*v*) and solvent B (ethyl acetate) was applied: 0–16 min, 0–60% B; 16–30 min, 60% B; 30–32 min 100% B and 32–35 min 100% A [74]. Carotenoids were identified using Chromeleon Chromatography Data System software (Version 6.3, ThermoFisher Scientific, Waltham, MA, USA) and quantified according to the calibration performed with 100 μL of neoxanthin, violaxanthin, lutein and β-carotene standards (Sigma-Aldrich, Lisboa, Portugal) (the same volume injected of each sample).

### 4.4. Biomass Growth and Protein Production (7 L Fermenter)

The selected mutant 8F4, cultivated initially in a 250-mL Erlenmeyer, was scaled-up to a 1-L Erlenmeyer flask and then to a 7-L benchtop fermenter (New Brunswick BioFlo^®^ CelliGen^®^115; Eppendorf AG, Hamburg, Germany) with an initial working volume of 3 L. This strain was also cultivated with HM medium at 30 °C. In addition, glucose (500 g L^−1^) was supplied in fed-batch mode to ensure a concentration range between 0.1 and 20 g L^−1^, and ammonia (24%) was used to keep a pH of 6.5. Non-limiting dissolved oxygen was guaranteed by keeping the airflow around 1 vvm and by increasing the agitation rate from 200 to 1200 rpm throughout growth.

Daily samples were collected aseptically to analyse the OD600 and/or DW, to measure the offline pH, to observe the culture in the microscopy and to store biomass, after centrifugation for 3 min at 2547× *g* (Hermle^®^ Z 300 centrifuge, Wehingen, Germany), to quantify protein content posteriorly.

### 4.5. Biostimulant Activity (In Vitro Assays)

The seed germination bioassay was used to determine the biostimulant activity of the F4 strain according to Zucconi et al. [75]. Garden cress (*Lepidium sativum* L., World of Flowers Sp., Poland) seeds were used as a model species.

Water suspensions of the dried biomass were prepared at 9 concentrations, namely, 0.01; 0.05; 0.1; 0.25; 0.5; 0.75; 1.0; 1.5 and 2.0 g L^−1^. Sterile deionised water was used as the negative control. Two positive controls were used, gibberellic acid (GA) at 0.00087 g L^−1^ and a commercial algae-based biostimulant product (Algaman B, Hubel Verde, Portugal) at 2.0 g L^−1^.

For each treatment, 5 replicates of 10 seeds were placed in Petri dishes with 2 Whatman No 1 filter papers, and 11 mL of either treatment was added. The seeds were placed in a growth chamber under controlled conditions of temperature (20 ℃) and ventilation (40%) in the dark and were left to germinate for 3 days. Afterwards, the length of the radicle and the young stem were measured for each seedling using a digital calliper (iGaging^®^ CoolantCal IP67, San Clemente, CA, USA).

Finally, the germination index (GI), relative growth of the radicle (RGR) and relative total growth (RTG) were calculated according to the following equations:(8)$$GI\left(\%\right)=\frac{\#\;germinated\;seeds\left(sample\right)\times radicle\;lenght(sample)}{Mean\;\#\;germinated\;seeds\left(negative\;control\right)\times Mean\;radicle\;lenght(negative\;control)}\times 100,$$
(9)$$RGR\left(\%\right)=\frac{radicle\;lenth\left(sample\right)}{Mean\;radicle\;lenth\left(neagtive\;control\right)}\times 100,$$
(10)$$RTG\left(\%\right)=\frac{Total\;lenght\;seedling\left(sample\right)}{Mean\;total\;lenght\;seedling\left(negatiove\;control\right)}\times 100$$

### 4.6. Statistical Analysis

All the experiments were carried out in biological triplicates, except the in vitro biostimulant activity assays, which were performed with five replicates. The results, expressed by mean ± standard deviation, were analysed by one-way Analysis of Variance (ANOVA), followed by Tukey’s HSD post hoc test, with a confidence interval of 95% (XLStat software, v2401.16.0, Microsoft^®^ Excel^®^).

## 5. Conclusions

This is the first study that described the usage of a high-throughput technology, fluorescence-activated cell sorting, to develop a pipeline to generate and select protein-rich mutants. The unavailability of appropriate and effective selection methodologies to isolate mutants with the desired target phenotypes is one of the most significant limitations of random mutagenesis. There are many strategies reported regarding the improvement in pigments, lipidic and carbohydrate contents of both wildtype and mutant microalgal strains for several biotechnological applications, but no work has been developed concerning protein content, which is one of the components of microalgal biomass with the greatest potential for agricultural and cosmetic applications. This work not only presents a novel selection strategy to target a combination of high protein and pigments contents, but also unveils the potential of the isolated mutant of *Chlorella vulgaris* F4 for an interesting application as a plant biostimulant. This mutant exhibited a 38% higher biomass productivity, 62% higher chlorophyll content and 19% higher protein content when compared to the wildtype (WT).

The biomass obtained from the scale-up of this strain enhanced the germination index and the relative total growth of garden cress seeds by 7% and 19%, respectively, when 0.1 g L^−1^ was applied, which highlight its biostimulant potential. As for future perspectives, it would be interesting to further analyse this biomass to understand which components provide this effect, such as the amino acid profile and the identification and quantification of phytohormones and their precursors. Additionally, downstream treatments of the biomass produced should also be investigated to further enhance the biostimulant capacity, namely by disrupting cells and releasing bioactive compounds. Finally, this scale-up process should be further optimised to achieve higher biomass and protein productivities, namely by optimising abiotic conditions, culture medium and cultivation and feeding strategies.

## Figures and Tables

**Figure 1 plants-13-02441-f001:**
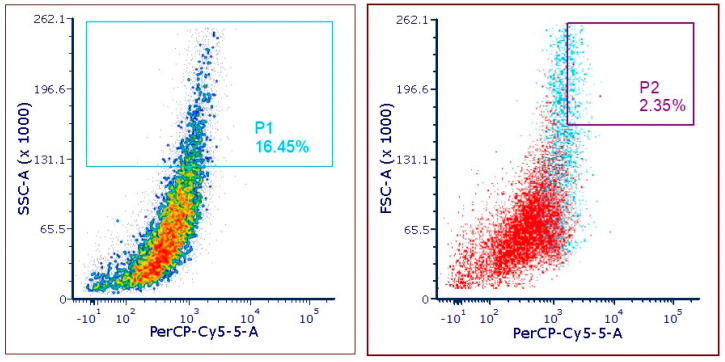
Fluorescence-activated cell sorting (FACS) procedure to isolate protein- and chlorophyll-rich mutants of *C. vulgaris*. The first gate, P1, was applied by combining the inner cell complexity (side scatter—SSC) and chlorophyll autofluorescence (PerCP-Cy5-5-A) and the other gate, P2, was applied by combining the cell size/volume (forward scatter—FSC) with chlorophyll autofluorescence (PerCP-Cy5-5-A).

**Figure 2 plants-13-02441-f002:**
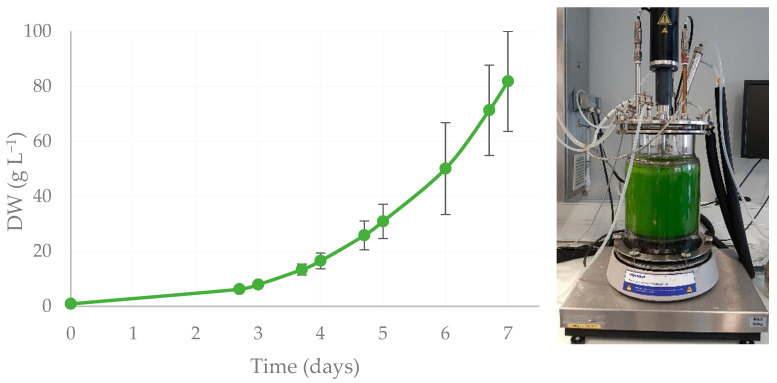
Growth curve of *C. vulgaris* mutant F4 in a 7-L reactor in heterotrophic conditions throughout 7 days. Data points on each day are displayed as mean ± SD (*n* = 3).

**Figure 3 plants-13-02441-f003:**
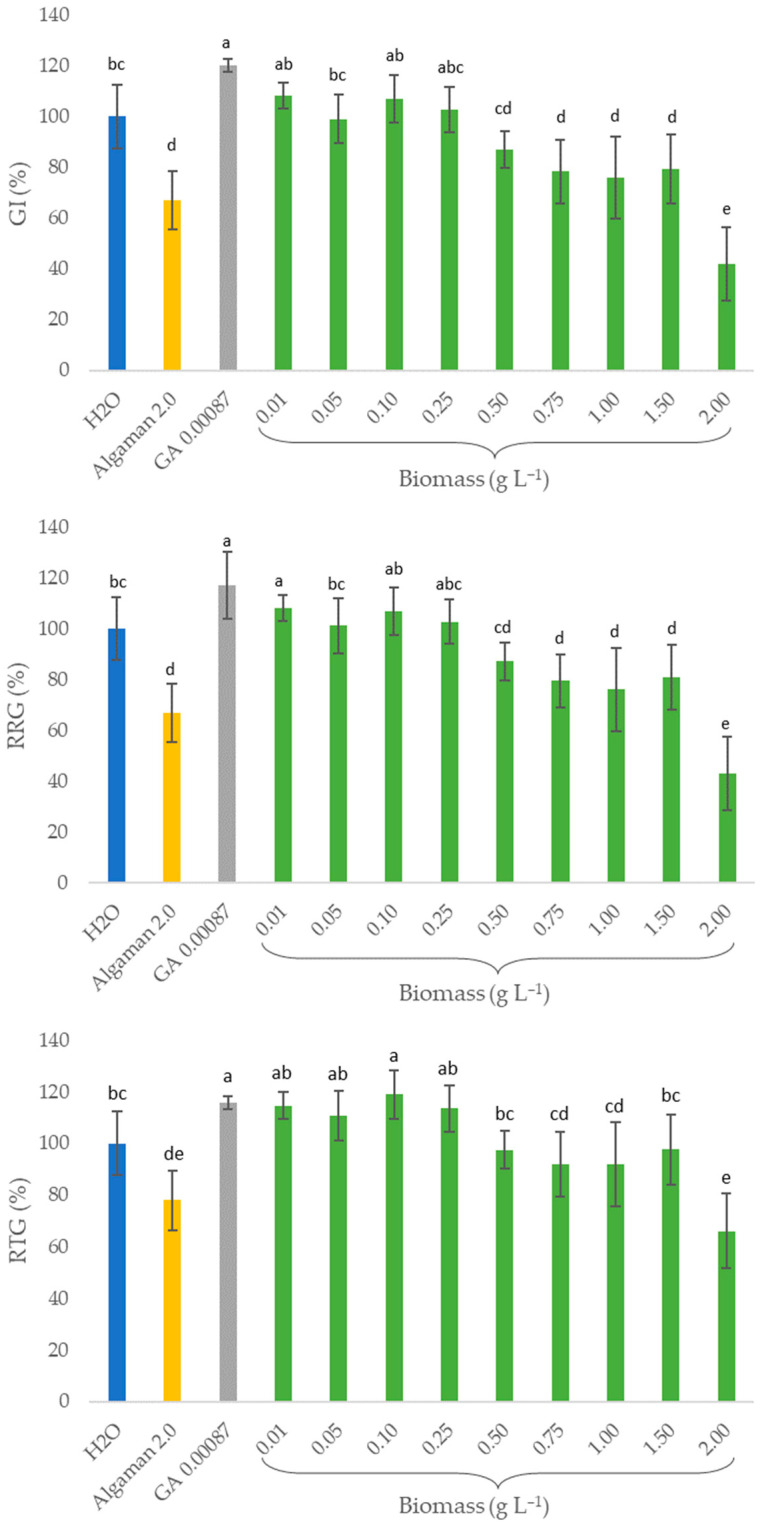
Germination index (GI, %); relative radicle growth (RRG, %); and total relative growth (RTG, %) for the *C. vulgaris* F4 strain at different concentrations: 0.01, 0.05, 0.1, 0.25, 0.5, 0.75, 1.0, 1.5 and 2.0 g L^−1^. H_2_O, sterile distilled water, was used as the negative control. Algaman and gibberellic acid (GA) were used as positive controls at the concentrations of 2.0 and 0.00087 g L^−1^, respectively. Bars represent the mean value ± SD, *n* = 5. Different letters indicate significant differences (*p* < 0.05) between treatments.

**Figure 4 plants-13-02441-f004:**
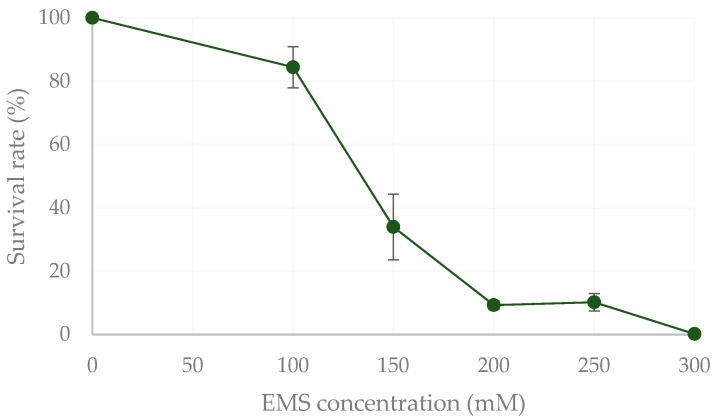
Survival rate (%) of *Chlorella vulgaris* 8WT exposed to different ethyl methanesulfonate (EMS) concentrations. Results are shown as mean ± SD, *n* = 3.

**Table 1 plants-13-02441-t001:** Biomass productivity (P) in g L^−1^ d^−1^, growth rate (µ) in d^−1^, protein content in % of dry weight (DW) and protein productivity (PP) in g L^−1^ d^−1^ of the wildtype (WT) and the 17 mutants of *C. vulgaris*, isolated by FACS.

Strain	P (g L^−1^ d^−1^)	µ (d^−1^)	Protein (% DW)	PP (g L^−1^ d^−1^)
WT	1.51 ± 0.05 ^b^	1.05 ± 0.02 ^d^	32.1 ± 1.1 ^c^	0.56 ± 0.05 ^b^
A6	1.54 ± 0.32 ^b^	1.05 ± 0.06 ^d^	31.8 ± 0.8 ^c,d^	0.51 ± 0.03 ^b,c,d^
B4	1.26 ± 0.02 ^b,e^	0.99 ± 0.01 ^c,d^	29.8 ± 1.1 ^c,d^	0.46 ± 0.02 ^b,c,d^
C1	1.39 ± 0.02 ^b,c^	1.05 ± 0.01 ^c^	29.9 ± 0.8 ^c,d^	0.47 ± 0.06 ^b,c,d^
C3	1.54 ± 0.05 ^b^	1.07 ± 0.02 ^c^	29.2 ± 0.7 ^c,d^	0.53 ± 0.02 ^b,d^
C4	1.52 ± 0.10 ^b^	1.04 ± 0.02 ^c^	31.0 ± 1.4 ^c,d^	0.68 ± 0.12 ^a,b,d^
C5	1.88 ± 0.13 ^a,b^	1.12 ± 0.02 ^b,c^	38.2 ± 0.4 ^b^	0.66 ± 0.01 ^a,b,d^
C10	1.49 ± 0.13 ^b^	1.07 ± 0.01 ^c^	30.8 ± 0.9 ^c,d^	0.50 ± 0.02 ^b,c,d^
D3	1.40 ± 0.03 ^b,d^	1.05 ± 0.01 ^c^	33.3 ± 0.4 ^c^	0.50 ± 0.01 ^b,c,d^
D4	1.35 ± 0.10 ^b,d^	1.05 ± 0.01 ^c^	30.3 ± 0.4 ^c,d^	0.46 ± 0.04 ^b,c,d^
D5	1.36 ± 0.08 ^b,d^	1.05 ± 0.02 ^c^	28.0 ± 0.7 ^c,d^	0.44 ± 0.02 ^b,c,d^
E2	0.84 ± 0.04 ^e^	0.95 ± 0.01 ^d^	39.1 ± 0.2 ^b^	0.43 ± 0.03 ^b,c,d^
E4	1.24 ± 0.15 ^b,d^	1.02 ± 0.04 ^c,d^	27.9 ± 1.5 ^c,d^	0.39 ± 0.01 ^d^
E5	1.22 ± 0.04 ^b,d^	1.02 ± 0.01 ^c,d^	30.3 ± 0.7 ^c,d^	0.43 ± 0.02 ^b,c,d^
F4	2.08 ± 0.17 ^a^	1.28 ± 0.05 ^a^	38.1 ± 0.5 ^b^	0.64 ± 0.03 ^a,b^
F5	1.88 ± 0.08 ^a,b^	1.26 ± 0.03 ^a^	38.2 ± 1.1 ^b^	0.63 ± 0.02 ^a,b^
G2	1.67 ± 0.18 ^b^	1.23 ± 0.01 ^a^	43.0 ± 1.7 ^a^	0.60 ± 0.03 ^a,b^
G3	1.55 ± 0.10 ^b^	1.05 ± 0.01 ^c^	35.6 ± 1.7 ^b,c^	0.60 ± 0.05 ^a,b^

Results are shown as mean ± SD, *n* = 3. Different letters indicate significant differences (*p* < 0.05) between strains. The most promising mutants and the WT are highlighted in green.

**Table 2 plants-13-02441-t002:** *Chlorella vulgaris* wildtype’s (WT) and F4 and G2 mutant strains’ chlorophyll content (mg g^−1^ of DW) as determined by Ritchie’s method and carotenoid concentrations (mg g^−1^ of DW) as determined by HPLC. n.d.—not detected; <LOQ—below limit of quantification.

	Chlorophyll *a*	Chlorophyll *b*	Total Chlorophyll	Neoxanthin	Violaxanthin	Lutein	*β*-Carotene
WT	0.30 ± 0.03 ^a^	0.18 ± 0.02 ^a^	0.48 ± 0.05 ^a^	0.36 ± 0.01 ^a^	0.17 ± 0.01	0.53 ± 0.01 ^a^	0.74 ± 0.02 ^a^
F4	0.54 ± 0.01 ^b^	0.24 ± 0.01 ^c^	0.78 ± 0.01 ^c^	0.20 ± 0.01 ^b^	<LOQ	0.59 ± 0.04 ^a^	0.37 ± 0.05 ^b^
G2	0.48 ± 0.03 ^b^	0.21 ± 0.01 ^e^	0.69 ± 0.04 ^c^	0.22 ± 0.03 ^b^	n.d.	0.82 ± 0.08 ^b^	0.67 ± 0.07 ^a^

Results are shown as mean ± SD, *n* = 3. Different letters indicate significant differences (*p* < 0.05) between strains.

## Data Availability

Data will be made available upon request.

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
