# Peer review of "Isolation and Selection of Protein-Rich Mutants of Chlorella vulgaris by Fluorescence-Activated Cell Sorting with Enhanced Biostimulant Activity to Germinate Garden Cress Seeds"

_plants, 2024, doi:10.3390/plants13172441_

Round 1

Reviewer 1 Report

Comments and Suggestions for Authors

Review on „Isolation and selection of protein-rich mutants of Chlorella vulgaris by Fluorescence-Activated Cell Sorting with enhanced biostimulant activity to germinate garden cress seeds”.

Generally, I think, the research topic is up-to-date work, very interesting, and brings new data for this topic. In addition, the topic of the study perfectly fits into the scope of the Plants, MDPI journal. The manuscript is formatted according to the rules and contains the necessary sections.

The manuscript is well-written. A large amount of work was involved in the study and the manuscript contains valuable results. It means that the authors have taken great care in designing experiments, collecting data, and analyzing results to ensure the reliability and validity of their findings. However, the manuscript needs some minor improvements and corrections.

Each abbreviation must define the first time they are used in each of three sections: the abstract; the main text; the first figure or table. Please check the Oxford comma throughout the whole manuscript.

Abstract:

Please rewrite the abstract. Present more information about the study. Approximately one-third of the abstract is an explanation of the background of the study. Please write only one sentence about the scientific background and emphasize the clear goal and hypothesis of this study. Also present the future importance of the research data was presented.

The abstract should focus on four main points: 1. The hypothesis, the goal, and/or the purpose of the research. 2. Brief methods: applied treatment and which parameters were measured. 3. The most highlighted results. 4. Brief conclusion of the study.

Line 28: What does WT mean? If somebody read only the abstract they do not have any idea about the abbreviations used in the manuscript. Please fully write of every abbreviation when they first appear.

Keywords:

Please arrange keywords in alphabetical order.

Results:

Table 1. Line 156: Please delete „Protein contents were determined by measuring total N in an elemental analyser” This belongs to the Materials and Methods section.

Line 156-158: „Results are shown as mean ± SD, n=3. Different letters indicate significant differences (p < 0.05) between strains.” Please present these sentences below the table.

Table 2: „Results are shown as mean ± SD, n=3. Different letters indicate significant differences (p < 0.05) between strains.” Please present these sentences below the table.

Discussion:

Please refer to the presented data using Table 1, Table 2, Figure 1, etc.

Materials and Methods:

Line 415-417: The description is only about the storage of samples and no words about the biomass analysis that was performed. Please add this information.

Author Response

Summary

Thank you for the review of this manuscript and for the helpful comments. Please find the detailed responses below and the corresponding revisions/corrections in track changes in the re-submitted file.

Comment #1

Each abbreviation must define the first time they are used in each of three sections: the abstract; the main text; the first figure or table. Please check the Oxford comma throughout the whole manuscript.

Reply:

We thank the reviewer for pointing out this error. All abbreviations were checked throughout the whole manuscript, and the alterations are marked under track changes.

Comment #2
Abstract: Please rewrite the abstract. Present more information about the study. Approximately one-third of the abstract is an explanation of the background of the study. Please write only one sentence about the scientific background and emphasize the clear goal and hypothesis of this study. Also present the future importance of the research data was presented.

The abstract should focus on four main points: 1. The hypothesis, the goal, and/or the purpose of the research. 2. Brief methods: applied treatment and which parameters were measured. 3. The most highlighted results. 4. Brief conclusion of the study. Line 28: What does WT mean? If somebody read only the abstract they do not have any idea about the abbreviations used in the manuscript. Please fully write of every abbreviation when they first appear.

Reply:

The authors agreed and the abstract was rewritten as suggested (under track changes mode). Additionally, the meaning of the abbreviation WT was added.

Comment #3

Keywords: Please arrange keywords in alphabetical order.

Reply:

The order was corrected according to the reviewer’s suggestion.

Comment #4

Results: Table 1. Line 156: Please delete „Protein contents were determined by measuring total N in an elemental analyser” This belongs to the Materials and Methods section. Line 156-158: „Results are shown as mean ± SD, n=3. Different letters indicate significant differences (p < 0.05) between strains.” Please present these sentences below the table. Table 2: „Results are shown as mean ± SD, n=3. Different letters indicate significant differences (p < 0.05) between strains.” Please present these sentences below the table.

Reply:

The suggested corrections were carried out in both tables (under track changes mode).

Comment #5

Discussion: Please refer to the presented data using Table 1, Table 2, Figure 1, etc.

Reply:

All the elements were referred to throughout the discussion section as suggested (under track changes mode).

Comment #6

Materials and Methods: Line 415-417: The description is only about the storage of samples and no words about the biomass analysis that was performed. Please add this information.

Reply:

The biomass analysis is described in the following sections. However, the analysis to be performed was mentioned in that phrase to clarify (line 427).

Reviewer 2 Report

Comments and Suggestions for Authors

Dear authors, "Isolation and selection of protein-rich mutants of Chlorella vulgaris by Fluorescence-Activated Cell Sorting with enhanced biostimulant activity to germinate garden cress seeds" is quite interesting and worth investigation. Please see some comments:

1- Please double-check formatting and grammar;

2- The introduction should be more concise;

3- Why did you use random mutation? There are great alternatives?

4- Why did you focus on proteins? The phytohormones are well-know for it. You may study the sinergetic effect of phytohormones and other C. vulgaris compunds.

5- Please do a deeper discussion on the extraction approach. It really affects the quality of extract;

6- Personally speaking, it is fundamental to do fine chemical identification (for instance HPLC and CG-Q-TOF (non-target);

Regards

Author Response

Summary

Thank you for the review of this manuscript and for the helpful comments. Please find the detailed responses below and the corresponding revisions/corrections in track changes in the re-submitted file.

Comment #1

Please double-check formatting and grammar;

Reply:

The manuscript formatting and grammar were re-checked.

Comment #2

The introduction should be more concise;

Reply:
As suggested, the authors made some alterations in this section, under track changes mode, to be more concise. Non-essential information and the respective references were deleted.

Comment #3
Why did you use random mutation? There are great alternatives?

Reply:

Random mutagenesis is a more ready-to-use technology with fast result-delivery. As mentioned in the introduction, there are other approaches, such as adaptive laboratory evolution (ALE) and genetic engineering (GE) methodologies. However, ALE is a very time-consuming and lengthy strategy, while GE is not only time-consuming but also more costly, requires more knowledge about the genetics of the organism and comes up with several production and commercialization hurdles, given the introduction of foreign genetic material.

Comment #4

Why did you focus on proteins? The phytohormones are well-know for it. You may study the sinergetic effect of phytohormones and other C. vulgaris compunds.

Reply:

This study focused on protein because amino acids can be more easily screened due to the implemented analytic methods in our laboratory that use minimum quantities of biomass, and the biostimulant potential of amino acids is also well described. Here, we managed to create a strategy to efficiently screen for hundreds of mutans, based on a high-throughput methodology. This strategy has never been reported before and having this type of method to select strains based on their protein content has much more applications than biostimulants, for example in food, feed and cosmetics, while phytohormones would be more limited to that. However, it would be interesting to develop a similar strategy to phytohormones. In this case we focused on protein but we intend to carry out further work, namely by studying the amino acids profiles of this strain as well as phytohormones contents, how to improve both and how does that impact the biostimulatory effect, which might give origin to another publication.

Comment #5
Please do a deeper discussion on the extraction approach. It really affects the quality of extract;

Reply:

A paragraph about this topic was added in the discussion section under track changes mode (lines 345-353).

Comment #6

Personally speaking, it is fundamental to do fine chemical identification (for instance HPLC and CG-Q-TOF (non-target);

Reply:

The authors agree that it would be interesting to carry out further chemical identification of the bioactive compounds that this novel strain might produce and that could impact biostimulant activity. As mentioned in the end of the discussion and conclusion sections, these are future perspectives to continue working on this project.

Round 2

Reviewer 2 Report

Comments and Suggestions for Authors

Dear authors, the manuscript has improved.

The discussion on extraction could be deeper discussed;

Personally speaking the fine extraction is fundamental for clear understand on it.

Regards

Author Response

The authors thank the reviewer for the helpful comments and revisions. The authors agree that it would be interesting to do a deeper discussion about this topic. However, it is not the focus of this manuscript and the discussion section is quite long already. For these reasons, we decided not to further extend the discussion of this point. Nonetheless, more detailed information about the preparation of the algal extracts of the references mentioned was added to the discussion (lines 303-318).
